# Evaluation of Commercial Concentration Methods for Microscopic Diagnosis of Protozoa and Helminths in Human Stool Samples in a Non-Endemic Area

**DOI:** 10.3390/microorganisms10061237

**Published:** 2022-06-16

**Authors:** Brice Autier, Jean-Pierre Gangneux, Florence Robert-Gangneux

**Affiliations:** Univ Rennes, CHU Rennes, Inserm, EHESP, Irset (Institut de Recherche en Santé Environnement Travail), UMR_S 1085, 35000 Rennes, France; brice.autier@univ-rennes1.fr (B.A.); jean-pierre.gangneux@univ-rennes1.fr (J.-P.G.)

**Keywords:** intestinal parasites, stool concentration, microscopy, diagnosis, protozoa, helminths

## Abstract

The diagnosis of intestinal parasitic infections still widely relies on microscopic examination of stools and requires reliable reagents and staff expertise. The ParaFlo^®^ assays (Eurobio Ingen) are ready-to-use concentration methods for parasite egg detection, and they could improve reagent traceability and ease of manipulation. Ninety-three stool samples were analyzed with the ParaFlo^®^ concentration methods and then compared with routine microscopic methods for protozoa and helminth detection: seventy-eight were analyzed with ParaFlo^®^ Bailenger and in-house Thebault or Bailenger concentrations, and fifty-five were analyzed with ParaFlo^®^DC and the in-house merthiolate-formalin diphasic concentration (DC) method. Fully concordant results were obtained for 75%, 70%, and 69% of samples when comparing ParaFlo^®^ DC and in-house DC, ParaFlo^®^ Bailenger and in-house Bailenger, and ParaFlo^®^ Bailenger and Thebault, respectively. The performances of the ParaFlo^®^ assays did not differ statistically from that obtained with their in-house counterparts (Bailenger and DC) for the detection of protozoa, but ParaFlo^®^ Bailenger performed significantly poorer than the Thebault method (*p* < 0.001). No statistical differences were observed between the commercial and in-house methods for helminth detection. These marketed concentration methods could be used in routine if combined with other techniques for protozoa detection.

## 1. Introduction

Intestinal parasitic infections rank first for neglected tropical diseases [1]. Helminthic infections are the most frequent etiologies, as it has been estimated that 1.45 billion people are infected with soil-transmitted helminths (hookworms, *Ascaris lumbricoides*, and *Trichuris trichiura*) [2], and more than 600 million and about 230 million people are infected with *Strongyloides stercoralis* [3] and *Schistosoma* spp., respectively [4]. These infections have been described to be associated with impaired nutritional status or cognitive development, and they are responsible for the highest loss of healthy life years, particularly hookworm infection [2]. Protozoa are also responsible for frequent digestive infections worldwide, and *Cryptosporidium* spp. and *Entamoeba histolytica* rank third and fourth among killer intestinal parasites, respectively [5]. While the prevalence of intestinal parasites has dramatically decreased in northern countries since decades, there is a renewed interest in the diagnosis of parasitic infections due to the increasing number of travelers and migrants from endemic countries.

Despite the increasing number of commercial multiplex PCR assays designed to detect the most frequent protozoan infections, microscopic examination of stools remains the reference method for the diagnosis of most intestinal parasites [6]. For optimal results, this approach requires the combination of a direct wet mount, several concentration techniques, and trained operators. Commercial concentration kits have been proposed for non-expert labs to improve handiness and standardization, and to facilitate egg visualization. The ParaFlo^®^ Bailenger AE assay (Eurobio Ingen) and ParaFlo^®^ diphasic concentration (DC) kits (Eurobio Ingen) are both ready-to-use CE-IVD diphasic methods for the detection of parasites in stool samples, using ethyl acetate and merthiolate-iodin-formalin (MIF), respectively. This study aimed to evaluate their performance in the detection of parasites in human stool samples by comparing them to in-house methods.

## 2. Materials and Methods

### 2.1. Clinical Samples

Ninety-three stool samples were prospectively included for comparative analysis between in-house and commercial concentration methods. In routine diagnosis, various concentration methods are used in our lab, depending on clinical signs and/or epidemiological data. In the absence of clinical signs (mainly in the setting of systematic screening for cooking staff), stool samples were analyzed using a direct wet mount and the Bailenger concentration method (acetic acid/acetate/ether concentration). In returning travelers and migrants, the analysis of stool samples included a direct wet mount examination and a diphasic concentration (DC), combined with either the Thebault concentration method or the Bailenger concentration method. Additionally, other methods for strongyloidiasis diagnosis, such as the Baermann funnel concentration method and coproculture, were performed, but they were not relevant for the present study; thus, they are not discussed. Depending on the remaining amount of stool material after routine techniques used for diagnostic purposes, samples were tested in parallel with one of the ParaFlo^®^ assays, i.e., ParaFlo^®^ Bailenger or ParaFlo^®^ DC (Eurobio Ingen, Chilly-Mazarin, France), or both.

### 2.2. In-House Concentration Methods

For the Bailenger concentration method [7], a nut-sized sample was suspended in 100 mL of acetyl-acetate buffer and left for 1 min. The suspension was then filtered through a sieve and divided into two conic tubes, and an equal volume of ether was added. After agitation and degassing, the tubes were centrifuged at 1100× *g* for 3 min. The supernatants were discarded, a drop of 0.9% NaCl was added, and the two pellets were examined under a light microscope.

For the diphasic concentration (DC) method, the same amount of stool was suspended in 40 mL of MIF solution [7] and sieved. Five milliliters was transferred into a conical tube and two milliliters of ether was added. After thorough mixing, the tube was left for 3 min, degassed, and then centrifuged at 1100× *g* for 3 min. The supernatant was discarded, and the whole pellet was examined under the microscope.

For the Thebault concentration method [7], a nut-sized sample was suspended in 100 mL of Thebault solution (0.2% trichloro-acetic acid, 10% formalin), sieved, and left for 1 min. The solution was then transferred into a separating funnel, and 100 mL of ether was added. After thorough mixing and degassing, the funnel was put on its bracket for 2–5 min. The bottom clear liquid was collected in two conical tubes and centrifuged at 520× *g* for 2 min. The two pellets were wet-mounted and examined under light microscopy.

### 2.3. Commercial Concentration Methods

Stool concentration was examined using ParaFlo^®^ assays following the manufacturer’s instructions. For the ParaFlo^®^ DC assay, 4 g of stool was suspended in a device pre-filled with 25 mL of merthiolate-formalin (or diphasic coloration base (DC)). After thorough mixing, 200 µL of iodinated Lugol solution was added, and the sample was left for 3 min after gentle, up-down homogenization. Then, 5 mL was collected and transferred into a conical tube, and 2.5 mL of ether was added. Specimens were then agitated and centrifuged at 200× *g* for 5 min, and the supernatant was discarded, yielding a small pellet of concentrate. The pellet was resuspended in 0.9% NaCl and observed under a light microscope. For ParaFlo^®^ Bailenger, the protocol was roughly similar, with the following modifications: (i) the DC solution was replaced by aceto-acetate buffer, and (ii) no Lugol solution was added.

### 2.4. Statistical Analysis

Results of ParaFlo^®^ Bailenger were compared to those of the in-house Bailenger and the Thebault methods. The results of ParaFlo^®^ DC were compared to those obtained with in-house DC. The results are expressed as numbers and %. For each comparison, the results are shown as concordant and discordant results. When appropriate, the performances were compared using Fisher’s exact test using GraphPad Prism software, v5.0 (San Diego, CA, USA).

## 3. Results

### 3.1. Samples and Techniques

Of the 93 stool samples included in the study, 23 samples could be analyzed with the in-house and commercial Bailenger methods. For 55 samples, the Thebault method was performed instead of the Bailenger concentration method; thus, this technique was compared to ParaFlo^®^ Bailenger. Another 55 samples could be analyzed with the in-house and ParaFlo^®^ diphasic concentration methods. Among the 93 samples, 5 were analyzed with both Bailenger methods and both DC methods, and 35 were analyzed using the Thebault, ParaFlo^®^ Bailenger, and both DC methods. The total number of samples tested with each concentration technique is depicted in Figure 1.

### 3.2. Overall Detection of Parasite Species

At least one parasite was detected in 59/93 samples using one of the techniques, and 34 samples tested negative with all methods. Fifteen parasite species (nine protozoa species and six helminth species) were detected with at least one technique (Table 1). The mean number of parasite species per sample was 1.56 ± 0.75. At first glance, the in-house methods (Bailenger or Thebault) appeared to be more efficient than the ParaFlo^®^ Bailenger assay in detecting protozoan parasites, but the results are not statistically significant. Protozoa cysts showed important morphological changes with ParaFlo^®^ Bailenger, preventing identification in three samples (scored negative).

### 3.3. Diagnostic Concordance between In-House and Commercial Assays

Fully concordant results were obtained for 75%, 70%, and 69% of samples when comparing ParaFlo^®^ DC and in-house DC, ParaFlo^®^ Bailenger and in-house Bailenger, and ParaFlo^®^ Bailenger and Thebault, respectively (Table 2). Partially discordant results (detection of some parasite species but missing some others) were observed for 5% and 18% of samples when comparing ParaFlo^®^ DC vs. in-house DC and ParaFlo^®^ Bailenger vs. Thebault, respectively. The highest number of all discrepant results was observed amongst the two Bailenger methods (30%, 7/23). Partially concordant results and false negative results were most often observed using the ParaFlo^®^ Bailenger assay compared to the Thebault method (Table 2).

### 3.4. Helminth and Protozoa Detection Performance

The performances of the ParaFlo^®^ assays were similar to those of their in-house counterparts (Bailenger and DC) for the detection of protozoan intestinal parasites (Table 3). However, ParaFlo^®^ Bailenger missed the detection of protozoa in five stool samples (*Giardia intestinalis* in two samples, *E. nana* in four, *E. coli* in one, and *Blastocystis hominis* in one), while *Sarcocystis hominis* was missed in one sample when using the in-house Bailenger method (*p* = 0.114). Compared to the in-house Thebault method, ParaFlo^®^ Bailenger yielded false negative results for 13 samples, of which 11 had multiple parasites detected with the in-house method (*p* < 0.001, Table 3). Among these 13 samples with false negative results, the commercial Bailenger method failed to detect *G. intestinalis* in 3 samples, *Chilomastix mesnilii* in 1, *Entamoeba histolytica/dispar* in 4, *E. coli* in 8, *E. hartmannii* in 4, *E. nana* in 5, and *B. hominis* in 1. Regarding the detection of helminth eggs, no statistical differences were observed between the commercial and in-house methods (Table 3). No bias of detection was observed in favor of one or the other helminth species when using the commercial and in-house DC techniques; *Schistosoma mansoni*, *Trichuris trichiura*, and *Enterobius vermicularis* were missed in one sample each when using the commercial DC method, while *S. mansoni*, *T. trichiura*, and *Hymenolepis nana* were missed in two, one, and one sample, respectively, when using the in-house DC method (data not shown).

## 4. Discussion

In this study, we compared two commercial concentration methods, the ParaFlo^®^ Bailenger and ParaFlo^®^ DC methods, to in-house concentration methods used as first-line techniques in our lab, i.e., the Thebault method, the Bailenger method, and the merthiolate-formol diphasic concentration method, for the detection of parasite elements in human stool samples. Overall, agreement between the marketed methods and in-house methods was moderate to good (69% to 75%). Of note, 30% of the discrepant results were observed in both Bailenger methods, mainly due to the lack of detection of protozoa, but our results did not reach statistical significance, which could be explained by the small number of stools tested (n = 25). Indeed, this technique is used in our lab only when a small number of stool samples are provided or are in systematic screening with no travel history. For protozoa detection, we prefer the Thebault technique, which indeed showed a much higher detection rate than that of the commercial Bailenger method (*p* < 0.001), as shown in Table 3. The Bailenger method and the closely related formalin-acetyl-acetate technique are widely used all over the world, and they are usually considered polyvalent techniques. Although our study was not designed to evaluate the pertinence of the choice of concentration techniques, it confirms that they do not perform equally for protozoa and helminth detection, and that two different techniques should be combined to improve diagnosis. By contrast to protozoa, the commercial and in-house assays produced similar results for helminth detection, regardless of the techniques compared. A limitation of our study was that we could not evaluate the use of all concentration methods on all samples, as it was limited by the remaining number of samples after routine examination. For this reason, we decided to compare the commercial assays to their in-house counterparts, one by one.

In the era of accreditation of clinical laboratories, the use of commercial diagnostic assays can seem attractive because of easy batch traceability and standardization of procedures compared to in-house reagents. However, it is important to verify that they have performances similar to those of routine techniques. Such evaluations are rare, as they require time, microscopic skills, and a sufficient number of stool samples to perform all methods in parallel, and this can be challenging in non-endemic countries. One recent study compared four commercial kits to a home-made procedure for the diagnosis of intestinal parasites [8]. The authors evaluated four commercial concentration methods (Easy Para Bailenger (Servibio), Mini Parasep Bailenger (Euro Bio), Paraprep S formalin (Euro Bio), and ELIstain Paratest/Para-Selles Bailenger (ELITechGroup)), and they observed that in-house concentration methods had the best performances. The performance depended on the parasite and the assay, which was expected, as the techniques were based on different physical and chemical processes. Another study reported that the mini-FLOTAC technique (MMS MedLab) was as efficient as a formol-ether concentration in-house method in detecting helminth eggs, but it performed poorly in detecting protozoa [9]. This discrepancy can also be attributed to the type of concentration method, as the authors compared a flotation method to a diphasic method. The mini-FLOTAC technique has been evaluated in several field studies in endemic countries, but it was often compared to the Kato–Katz method, which is rarely used in northern countries [10,11].

Nowadays, the use of microscopic techniques could appear outdated, given the increasing availability of molecular tools [12,13]. In fact, several multiplex PCR assays have been developed for protozoa detection with performances equal or superior to those of microscopy [14,15,16], and they are able to detect up to six different targets [17,18]. However regarding helminth detection, commercial multiplex PCR assays have not yet surpassed microscopy in terms of performance [19,20] due to several reasons: (i) the number of parasite eggs spread in feces is usually lower than that in the case of protozoa; (ii) DNA extraction for helminth PCR examination requires adapted mechanical pretreatment to ensure wall disruption without degrading DNA [21]; (iii) the spectrum of human pathogenic helminths is much wider than that of pathogenic protozoa, which makes it difficult to design a multiplex PCR assay covering all infections. This implies that microscopic examination of stool samples should be maintained, at least in reference centers. For non-expert laboratories, the use of a multiplex PCR assay for protozoa detection and a commercial concentration method for helminth detection could be a valuable and simple combined approach. The latter could be, for example, ParaFlo^®^ DC, which showed a performance similar to that of the DC in-house method in our hands. By restricting microscopic methods to helminth detection, this strategy would save time, as it allows one to reduce the number of concentration methods used and to examine mounted slides under a microscope at 10-fold magnification, which is rapid and easy. However, it should be remembered that *Strongyloides stercoralis* larvae are not concentrated by diphasic methods, and adequate methods or serological techniques should be combined when necessary [22,23].

What is the future for helminth detection? Many efforts have been made in the development of molecular methods to detect parasites in stool samples, initially by immunodetection (coproantigens) and then by molecular biology, but microscopic techniques are still unmatched for many helminths. Thus, the place of microscopy should probably be reconsidered, and efforts should focus on how to improve its ease of use. A promising approach could be the development of machine learning tools, which would save time through the automation of microscopic observation [24].

## 5. Conclusions

In-house methods showed better performances than ParaFlo^®^ concentration methods for protozoa detection. However, ParaFlo^®^ assays showed equivalent results for helminth detection. This suggests that these marketed concentration methods could be used in routine, provided that they are combined with molecular techniques for protozoa detection.

## Figures and Tables

**Figure 1 microorganisms-10-01237-f001:**
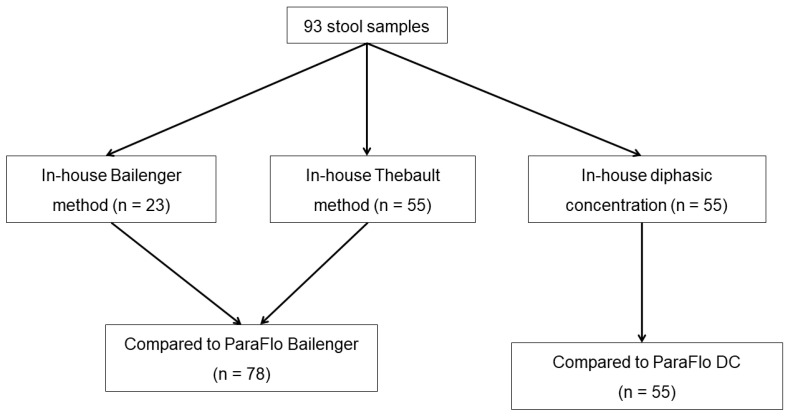
Flowchart of the study design and number of samples analysed.

**Table 1 microorganisms-10-01237-t001:** Parasite species detected using the in-house and concentration methods (n = 93 samples).

Parasite Species	Number of Positive Samples Detected With			
	In-House Bailenger or Thebault	ParaFlo^®^ Bailenger	In-House DC	ParaFlo^®^ DC
*Entamoeba coli*	19	13	4	6
*Endolimax nana*	19	12	4	1
*Entamoeba histolytica/dispar*	4	0	0	1
*Entamoeba hartmanni*	4	1	1	0
*Blastocystis hominis*	2	0	0	0
*Cryptosporidium* sp.	1	1	0	0
*Sarcocystis hominis*	0	1	0	0
*Giardia intestinalis*	12	7	2	1
*Chilomastix mesnilii*	2	2	0	0
*Schistosoma mansoni*	5	6	12	13
*Hymenolepis nana*	2	3	3	4
*Enterobius vermicularis*	0	0	1	0
Ancylostomatidae	1	1	0	0
*Trichuris trichiura*	2	2	1	1
*Taenia* sp.	1	1	0	0
Total	74	50	28	27

**Table 2 microorganisms-10-01237-t002:** Detailed results of in-house methods and ParaFlo^®^ methods, compared two by two.

No. of Samples With	ParaFlo^®^ Bailenger vs. In-House Bailenger N = 23	ParaFlo^®^ Bailenger vs. Thebault Method N = 55	ParaFlo^®^DC vs. In-House DC N = 55
Concordant results	16 (70)	38 (69)	41 (75)
Concordant negative results, n (%)	10 (43.5)	18 (33)	29 (52.7)
Concordant positive results, n (%)	6 (26)	20 (36)	12 (21.8)
Partially positive results of ParaFlo^®^ compared to in-house method, n (%)	0	8 (15)	2 (3.5)
Partially positive results of in-house method compared to ParaFlo^®^, n (%)	0	2 (3)	1 (2)
False negative result of ParaFlo^®^ method, n (%)	6 (26)	6 * (11)	6 (11)
False negative result of in-house method, n (%)	1 (4.5)	2 * (3)	5 (9)

vs., versus; *one sample was counted in both categories, as a different parasite was detected with each technique.

**Table 3 microorganisms-10-01237-t003:** Separate analysis of the performances of commercial techniques compared to routine procedures for the detection of protozoa and helminths.

Comparison of Techniques	Overall Concordance % (n/N)	No. of Samples with False Negative Results Using:		*p*-Value ^1^
		The Commercial Technique	The In-House Technique	
Detection of protozoa				
ParaFlo*^®^* Bailenger vs. in-house Bailenger	74% (17/23)	5	1	0.114 ns
ParaFlo*^®^* Bailenger vs. Thebault method	75% (41/55)	13	1	<0.001
ParaFlo*^®^* DC vs. in-house DC	85% (47/55)	5	3	0.65 ns
Detection of helminths				
ParaFlo*^®^* Bailenger vs. in-house Bailenger	96% (22/23)	1	0	1 ns
ParaFlo*^®^* Bailenger vs. Thebault method	93% (51/55)	1	3	0.586 ns
ParaFlo*^®^* DC vs. in-house DC	87% (48/55)	3	4	1 ns

^1^ Fisher’s exact test, ns; not significant.

## Data Availability

Data are available following the link https://drive.google.com/drive/folders/1ilpl35Vugr6BlvDTRVq_i2z8F3rJ7VA4?usp=sharing (accessed on 24 May 2022).

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
