# Peer review of "Evaluation of Commercial Concentration Methods for Microscopic Diagnosis of Protozoa and Helminths in Human Stool Samples in a Non-Endemic Area"

_microorganisms, 2022, doi:10.3390/microorganisms10061237_

Round 1

Reviewer 1 Report

The article „Evaluation of commercial concentration methods for microscopic diagnosis of protozoa and helminths in human stool samples in a non-endemic area” performs results from the studies of concentration methoods comparison to in-house methods. with commercial techniqes.

In my opinion the article provides valuable information, however it is difficult to read. Maybe it could be simplified a little to make it easier? I mean there is no information why not all 93 samples were tested by all methods? It should be explain in Discussion. The authors should also start the study with spiked samples with parasites ( a suitable, known numer) do the authors have such study results?

Furthermore, here are the points that should be considered/corrected

1.       Please include the reference in line 43.

2.       In M&M, section „clinical samples” there is no information on the number of samples, please add its here.

Author Response

The article „Evaluation of commercial concentration methods for microscopic diagnosis of protozoa and helminths in human stool samples in a non-endemic area” performs results from the studies of concentration methoods comparison to in-house methods. with commercial techniques.

In my opinion the article provides valuable information, however it is difficult to read. Maybe it could be simplified a little to make it easier? I mean there is no information why not all 93 samples were tested by all methods? It should be explain in Discussion.

Response: An additional information has been added in the discussion section (L179-183). Actually, the remaining quantity of stools was not sufficient to perform all techniques simultaneously.

The authors should also start the study with spiked samples with parasites ( a suitable, known numer) do the authors have such study results?

Response: This study is a comparison between commercial and in-house techniques on routine samples. If we agree that evaluatiing of the detection limit of the different methods could be of interest, this needs to perform protozoa culture (not done in our lab) in order to have a well-determined inoculum. For this reason, we are not able to realize spiked samples.

Furthermore, here are the points that should be considered/corrected

  1. Please include the reference in line 43.

Response: A reference emphasizing the importance of microscopic examination of stool samples has been added.

  1. In M&M, section „clinical samples” there is no information on the number of samples, please add its here.

Response: The total number of included stool samples was specified L54.

Reviewer 2 Report

Comparing the diagnostic efficacy of different coproparasitological methods is a hotly debated topic, with multiple published articles addressing this topic.

It is a correctly written article based on factual and accurate data.

Some minor spelling mistakes (e.g., correct article usage, conciseness of the phrasing, use of commas) must be corrected.

Example of the minor correction required - Line 62: "funnel" instead of "funel";

Author Response

We thank the reviewer for his comments.

The text has been reviewed to fix typical errors

Reviewer 3 Report

The authors proceeded to the comparison of different methodologies for protozoa detection.They compared in-house methodologies withParaFlo® concentration methods . The results showed equivalent results for helminths detection between the methodologies, but concerning protozoa detection the in-house methods showed better performances than ParaFlo® concentration methods.

The paper is well written , bibliography is up to date and methodology correct while the sample was important in order to permit statistical comparisons.

It is an article that should be of high interest to parasitologists and microbiologists.

my suggestion is TO ACCEPT and publish it in its present form

Author Response

We thank the reviewer for his comments. Typo errors have been fixed.